# Omics Sequencing of *Saccharomyces cerevisiae* Strain with Improved Capacity for Ethanol Production

Zhilong Lu [1,†], Ling Guo [2,†], Xiaoling Chen [1], Qi Lu [1], Yanling Wu [1], Dong Chen [1], Renzhi Wu [1,3] and Ying Chen [1,*]

1. National Key Laboratory of Non-food Biomass Technology, Guangxi Academy of Sciences, Nanning 530007, China
2. CAS Key Laboratory of Environmental and Applied Microbiology, Environmental Microbiology Key Laboratory of Sichuan Province, Chengdu Institute of Biology, Chinese Academy of Sciences, Chengdu 610041, China
3. Guangxi Key Laboratory of Polysaccharide Materials and Modification, School of Marine Sciences and Biotechnology, Guangxi Minzu University, 158 West Daxue Road, Nanning 530008, China
* Correspondence: cy2008-5-1@163.com
† These authors contributed equally to this work.

**Abstract:** *Saccharomyces cerevisiae* is the most important industrial microorganism used to fuel ethanol production worldwide. Herein, we obtained a mutant *S. cerevisiae* strain with improved capacity for ethanol fermentation, from 13.72% ($v/v$ for the wild-type strain) to 16.13% ($v/v$ for the mutant strain), and analyzed its genomic structure and gene expression changes. Kyoto Encyclopedia of Genes and Genomes (KEGG) enrichment revealed that the changed genes were mainly enriched in the pathways of carbohydrate metabolism, amino acid metabolism, metabolism of cofactors and vitamins, and lipid metabolism. The gene expression trends of the two strains were recorded during fermentation to create a timeline. Venn diagram analysis revealed exclusive genes in the mutant strain. KEGG enrichment of these genes showed upregulation of genes involved in sugar metabolism, mitogen-activated protein kinase pathway, fatty acid and amino acid degradation, and downregulation of genes involved in oxidative phosphorylation, ribosome, fatty acid and amino acid biogenesis. Protein interaction analysis of these genes showed that glucose-6-phosphate isomerase 1, signal peptidase complex subunit 3, 6-phosphofructokinase 2, and trifunctional aldehyde reductase were the major hub genes in the network, linking pathways together. These findings provide new insights into the adaptive metabolism of *S. cerevisiae* for ethanol production and a framework for the construction of engineered strains of *S. cerevisiae* with excellent ethanol fermentation capacity.

**Keywords:** *Saccharomyces cerevisiae*; ethanol fermentation; omics sequencing; expression trend; protein interaction; hub genes

## 1. Introduction

Ethanol is a potential environmentally friendly alternative to fossil fuels that can be used to propel light vehicles with gasoline, thereby improving octane numbers and reducing environmental pollution [1]. In addition, it is the premier biotechnological global product in terms of volume and economic value [2,3]. Sucrose is an abundant, cheap, and readily available substrate for industrial fermentation, and its use in the production of fuel ethanol has proved successful in Brazil [4,5]. Sugarcane contains 11–18% (wet $W/W$) sugars, comprised of 90% sucrose and 10% glucose and fructose [6]. During the edible sugar-making process, cane molasses is generated as a by-product in vast amounts, containing 45–60% sucrose and ~5–20% glucose and fructose [5,7]. Sucrose production is the most important industry in the Guangxi Zhuang Autonomous Region, accounting for ~60% of the total planting area and yield of sugarcane in China.

Sucrose-based fuel ethanol production is one of the world's major biotechnological processes, which utilizes the microorganism *Saccharomyces cerevisiae* due to its ability to efficiently ferment sugars to ethanol and resist industrial environmental stresses [8,9]. Strains with optimal ethanol fermentation capacities can improve the efficiency of bioethanol production and reduce energy consumption and production costs considerably [3]. Thus, various strategies have been employed with the aim of developing new strains and improving ethanol production, ranging from early hybridization, protoplast fusion, and mutagenesis to recent genetic engineering, gene editing, and synthetic biology [10–12].

In the fermentation process, ethanol production, carbon dioxide generation, redox balancing, energy production, and cell growth are all coupled. Although the physiological and biochemical processes of yeast ethanol fermentation have been well characterized, there remains much to learn about the relationship between gene expression and cell characteristics [13]. Omics sequencing is a powerful tool that can be used to explore the relationship between gene mutations, gene expression differences, and cell trait changes to better understand the principles underlying physiological homeostasis and potentially uncover additional molecular phenotypes associated with specific characteristics [14–18].

In this study, two strains of *S. cerevisiae* were characterized, a wild type (WT) and a mutant (MT) strain, with an obvious advantage over the WT strain in ethanol fermentation when grown in a 30% (*w/v*) sucrose medium. Whole genome sequencing (WGS) and RNA sequencing (RNA-Seq) were used to analyze genomic differences and gene expression trends during the fermentation process between the two strains. Venn diagram analysis, KEGG enrichment analysis, and protein interaction analysis were used to characterize changes in gene expression in the MT strain during fermentation and identify key hub genes of metabolic adaptability. This study will help to better understand the metabolic characteristics of *S. cerevisiae* with a high capacity for ethanol fermentation and provide ideas for the construction of engineered *S. cerevisiae*.

## 2. Materials and Methods

### 2.1. Strains and Culture Conditions

*S. cerevisiae* WT (MATa/MATα) (CGMCC 2.4748) [19] is a wild-type diploid strain isolated from year-old sugar mill waste in Nanning, China, and *S. cerevisiae* MT (MATa/MATα) is a mutant of the WT strain. The MT strain was obtained by random mutation using UV irradiation; the screening process was as follows: the mutated cell solution was coated on a culture dish (150 mm) with a yeast peptone dextrose (YPD) medium (20 g L$^{-1}$ tryptone, 10 g L$^{-1}$ yeast extract, and 20 g L$^{-1}$ glucose), cultured at 30 °C for two days. The grown cell moss on the culture dish was evenly divided into 16 regions, and each was scraped into 5 mL YPD medium for overnight culture at 30 °C, then 100 uL cell solution was absorbed into 5 mL YPD medium for overnight culture. Cell concentrations were measured, comparative amounts of cells were inoculated into 100 mL YPS30 medium (20 g L$^{-1}$ tryptone, 10 g L$^{-1}$ yeast extract, and 300 g L$^{-1}$ sucrose) in 250 mL Erlenmeyer flasks, and ethanol fermentation was allowed to proceed at 30 °C, 180 rpm. Cells with the highest ethanol yield were coated on a culture dish again, repeated the above process five times, then isolated single colonies from the stain with the highest ethanol yield. After two rounds of isolation, we got the purified mutant with the highest ethanol yield. When the fermentation experiment was performed in this paper, cells were cultivated in a YPD medium at 30 °C, 180 rpm. Comparative amounts of freshly cultured WT and MT cells were inoculated into a 100 mL YPS30 medium in 250 mL Erlenmeyer flasks, and ethanol fermentation was allowed to proceed at 30 °C, 180 rpm.

## 2.2. Detection of the Fermentation Process

The number of cells was counted using an automated cell counter (IY1200 Counstar, Ruiyu, Shanghai, China) every 4 h. Ethanol production was quantified using a gas chromatograph (6890, Agilent, Palo Alto, CA, USA). In the fermentation broth, reducing sugar was determined using the dinitrosalicylic acid response. Following HCl hydrolysis, the total residual sugar content of the broth was quantified using the dinitrosalicylic acid response every 8 h. RNA-Seq was used to analyze gene expression at 16 h (T1), 40 h (T2), and 64 h (T3) during fermentation. Three parallel experiments were performed for each sample.

## 2.3. Whole Genome DNA Extraction, Library Construction, and Sequencing

Total genomic DNA was obtained by completely grinding cells in liquid nitrogen, decontaminating, adding cetyltrimethylamine bromide to facilitate the removal of polysaccharides, deproteinizing with phenol and chloroform, precipitating with isopropanol and absolute ethyl alcohol, washing with 75% alcohol to remove the sediment, and finally dissolving the pellet in ddH$_2$O. DNA quality was assessed using a Nanodrop Microspectrophotometer (Nanodrop 2000, Thermo Fisher Scientific, Waltham, MA, USA) and agarose gel electrophoresis. At least 3 µg of genomic DNA was used to construct paired-end libraries with an insert size of 500 bp using a Paired-end DNA Sample Prep kit (Illumina Inc., San Diego, CA, USA). These libraries were then sequenced on a NovaSeq6000 system using a PE 150 strategy by GeneDenovo Biotechnology Co., Ltd. (Guangzhou, China).

## 2.4. RNA Extraction, Library Construction, and Sequencing

Total RNA was extracted using a Trizol reagent kit (Invitrogen, Carlsbad, CA, USA) according to the manufacturer's protocol. RNA quality was assessed on an Agilent 2100 Bioanalyzer (Agilent Technologies, Palo Alto, CA, USA) and checked using RNase-free agarose gel electrophoresis. mRNA was enriched using Oligo (dT) beads. The enriched mRNA was fragmented into short fragments using a fragmentation buffer and reverse transcribed into complementary DNA (cDNA) using a NEB_Next Ultra RNA Library Prep Kit for Illumina (NEB #7530, New England Biolabs, Ipswich, MA, USA). The purified double-stranded cDNA fragments were end-repaired, followed by the addition of an adenylate (A) base; then, the fragments were ligated to Illumina sequencing adapters. The ligation reaction product was purified using AMPure XP Beads (1.0×). The ligated fragments were subjected to size selection using agarose gel electrophoresis and amplified using polymerase chain reaction (PCR). The resulting cDNA libraries were sequenced using an Illumina NovaSeq 6000 by GeneDenovo Biotechnology Co., Ltd (Guangzhou, China).

## 2.5. Sequence Data Analysis

The WGS and RNA-Seq raw reads were deposited in the Sequence Read Archive database with accession number PRJNA885247.

Raw WGS reads were processed to obtain high-quality clean reads by removing reads with $\geq$10% unidentified nucleotides (N), reads with >50% bases having Phred quality scores $\leq$ 20, and reads aligned to the barcode adapter. To identify single nucleotide polymorphisms (SNPs) and insertion–deletion mutations (InDels), the Burrows–Wheeler Aligner was used to align the clean reads from each sample against the reference genome (Ensembl_release100 of *S. cerevisiae*) [20]. Variant calling was performed for all samples using the GATK Unified Genotyper. SNPs and InDels were filtered using GATK's Variant Filtration with proper standards [21]. To determine the physical positions of each SNP, the software tool ANNOVAR was used to align and annotate SNPs or InDels [22]. Structural variation [23] was determined using the software BreakDancer (Max1.1.2., Ken Chen, The Genome Center, St. Louis, MO, USA) [24].

Raw RNA-Seq reads were further filtered using fastp (version 0.18.0) [25]. Reads containing adapters, $\geq$10% of unknown nucleotides, low-quality reads containing $\geq$50% low-quality (Q-value $\leq$ 20) bases, and reads mapped to rRNA were removed. The remaining clean reads were further used in assembly and gene abundance calculations. Using En-

sembl_release100 of the *S. cerevisiae* genome FASTA file as the reference genome, the index of the reference genome was built, and paired-end clean reads were mapped to the reference genome using HISAT2.2.4 [26]. The fragment per kilobase of transcript per million mapped reads (FPKM) value was calculated to quantify gene expression abundance and variations using RSEM software (v1.3.1, Bo Li, Departmet of Comuter Sciences, University of Wisconsin-Madison, Madison, WI, USA) [27]. Correlation analysis of two parallel experiments was performed using R Correlation. Gene differential expression analysis was performed using DESeq2 [28]. Gene expression pattern analysis was used to cluster genes of similar expression patterns from multiple samples (at least three in a specific time point, space, or treatment dose size order). To examine the expression pattern of differentially expressed genes (DEGs), the expression data of each sample (in the order of treatment) were normalized to 0, log2(v1/v0), and log2(v2/v0), and then clustered using Short Time-series Expression Mine software (STEM, Jason Ernst, School of Computer Science, Carnegie Mellon University, Pittsburgh, PA, USA) [29]. The clustered profiles with *p*-values $\leq 0.05$ were considered significant profiles. KEGG enrichment analysis of the target DEGs was performed using the KOBAS 3.0 package. The protein interaction network was analyzed using the interaction relationships in the STRING protein interaction database (http://string-db.org, accessed on 28 February 2022) [30], and the interaction network diagram was constructed using Cytoscape [31].

### 2.6. Quantitative Reverse Transcription PCR (qRT-PCR)

Two comparison groups, MT-T1 vs. MT-T2 and WT-T2 vs. MT-T2 were selected as objects, and qRT-PCR of 21 genes from the two comparison groups were used to verify the RNA-Seq accuracy. The amplified primers were shown in Table S1. Biological replicates of each gene were identical to those used for RNA-Seq. Reverse transcription was performed using the HiScrip II Q RT SuperMix kit for qPCR (R223, Vazyme, Nanjing, China), and qPCR was performed using the ChanQ SYBR qPCR Master Mix kit (Q341, Vazyme, Nanjing, China) on a TianLong 988 system (Tianlong, Xi'an, China). Data were analyzed using the $2^{-\Delta\Delta Ct}$ method, with bifunctional DRAP deaminase/tRNA pseudouridine synthase *RIB2* as the reference gene.

## 3. Results

### 3.1. Cell Growth and Fermentation

The *S. cerevisiae* MT strain had obvious advantages in growth and ethanol fermentation in 30% (*w/v*) sucrose medium compared to that of the WT strain. During the cell growth phase, the difference in cell number between the MT and WT strains was pronounced after 20 h, with a maximum mean cell number for the two strains of $3.06 \times 10^8$ and $3.23 \times 10^8$, respectively (Figure 1A). The trend in sugar consumption was consistent with that of ethanol yield; the difference between the two strains was shown after 24 h, and the ethanol yields were the highest at 64 h (13.72% *v/v* for the WT strain and 16.13% *v/v* for the MT strain (Figure 1B)). The total residual sugar content in the fermentation broth was stable from 48 h, while at the end of 80 h fermentation, the total residual sugar content was 41.52 g/L (WT) and 34.03 g/L (MT) (Figure 1C). The reduced sugar content in the fermentation broth showed an initial increase and subsequently decreased, peaking at 32 h. The sugar reduction of MT was significantly lower than that of WT after 16 h, at 31.0667 g/L (MT) and 41.5 g/L (WT) after 80 h fermentation (Figure 1D). Almost all the sucrose in the fermentation broth of the WT strain became reducing sugar.

### 3.2. Genome Sequencing

Strict quality control and data filtering were performed on the original data to obtain the high-quality clean reads used in the data analysis. Genome coverage, sequencing depth, SNPs, and InDels in the two samples were analyzed by aligning the sequences with the reference genome of *S. cerevisiae* (Ensembl_release100). The statistics on quality of sequencing data were shown in Tables S2–S5, on information of SNPs and InDels were shown in

Tables S6–S10. There was no obvious difference in the genome location of SNPs between the two strains, but SNP coding information in the MT strain contained significant changes compared to that of the WT; synonymous single-nucleotide variants (SNVs) increased, while nonsynonymous SNVs decreased (Figure 2A). The location of InDels in the MT genome changed in both the exons and upstream of the coding region, while the coding information changed in frameshift and nonframeshift deletions (Figure 2B). SV included translocation breakends (BND), deletions (DEL), tandem duplications (DUP), insertions (INS), and inversions (INV), shown in Figure 2C. Compared to that of the WT strain, the proportion of BNDs decreased, and INSs increased in the MT strain. The distribution density of SNPs on each chromosome was relatively high (Figure 3A), while the distribution density of InDels was high only on mitochondria (Figure 3B). The GO enrichment of the SNPs and InDels in MT strain was significantly enriched in cellular process, single-organism process, metabolic process (Biological Process), cell, cell part, and organelle (Cellular Component), binding and catalytic activity (Molecular Function) (Figure 4). The KEGG enrichment result of SNPs was basically consistent with the result of InDels (Figure 5). Significantly enriched metabolic pathways include carbohydrate metabolism, amino acid metabolism, metabolism of cofactors and vitamins, and lipid metabolism.

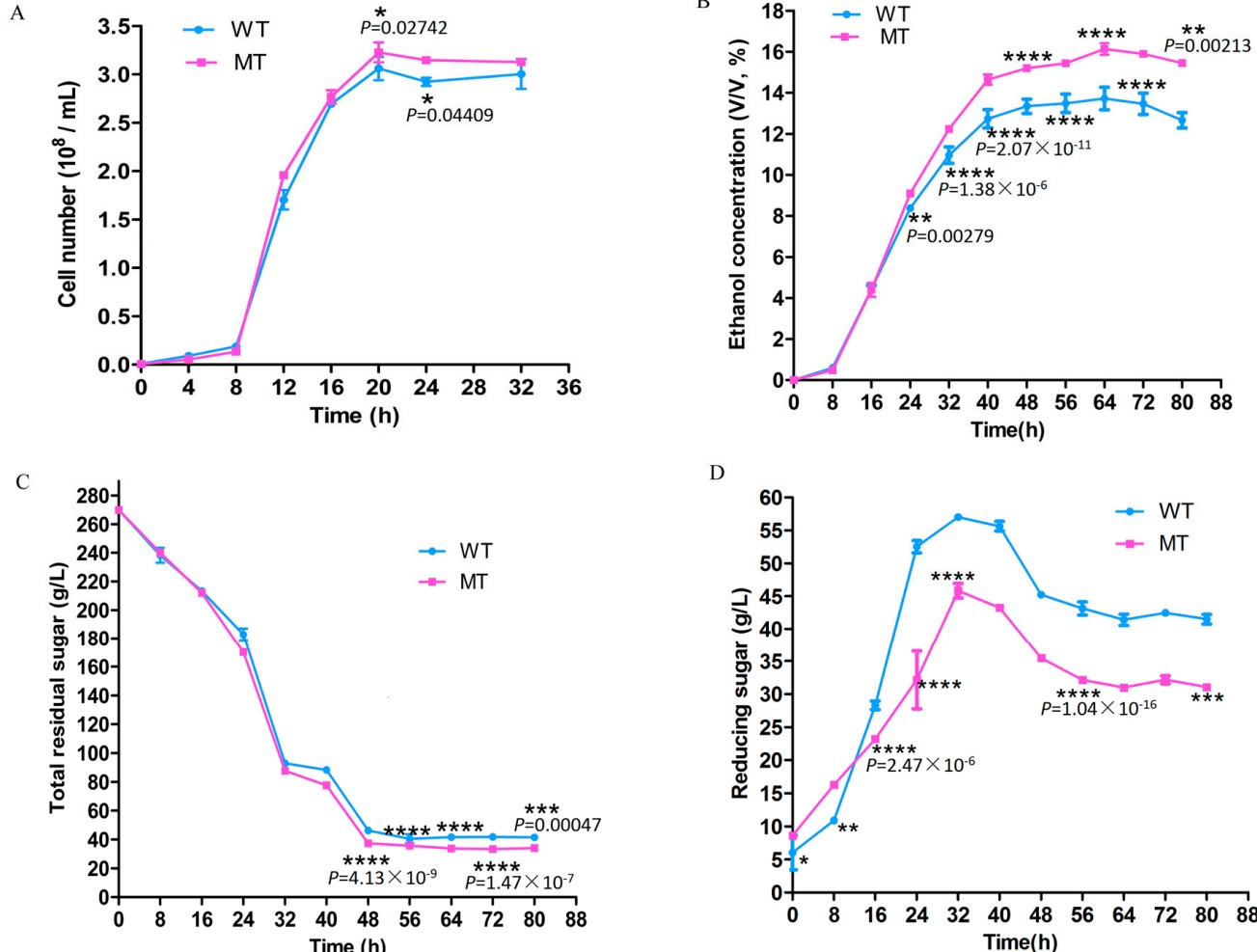

**Figure 1.** Cell growth and fermentation in YPS30 medium. Cell growth curves (**A**), ethanol yield curves (**B**), the total residual sugar content curves (**C**), and the reducing sugar content curves (**D**) of two strains. (Comparison of two strains * $p < 0.05$, ** $p < 0.01$, *** $p < 0.001$, and **** $p < 0.0001$).

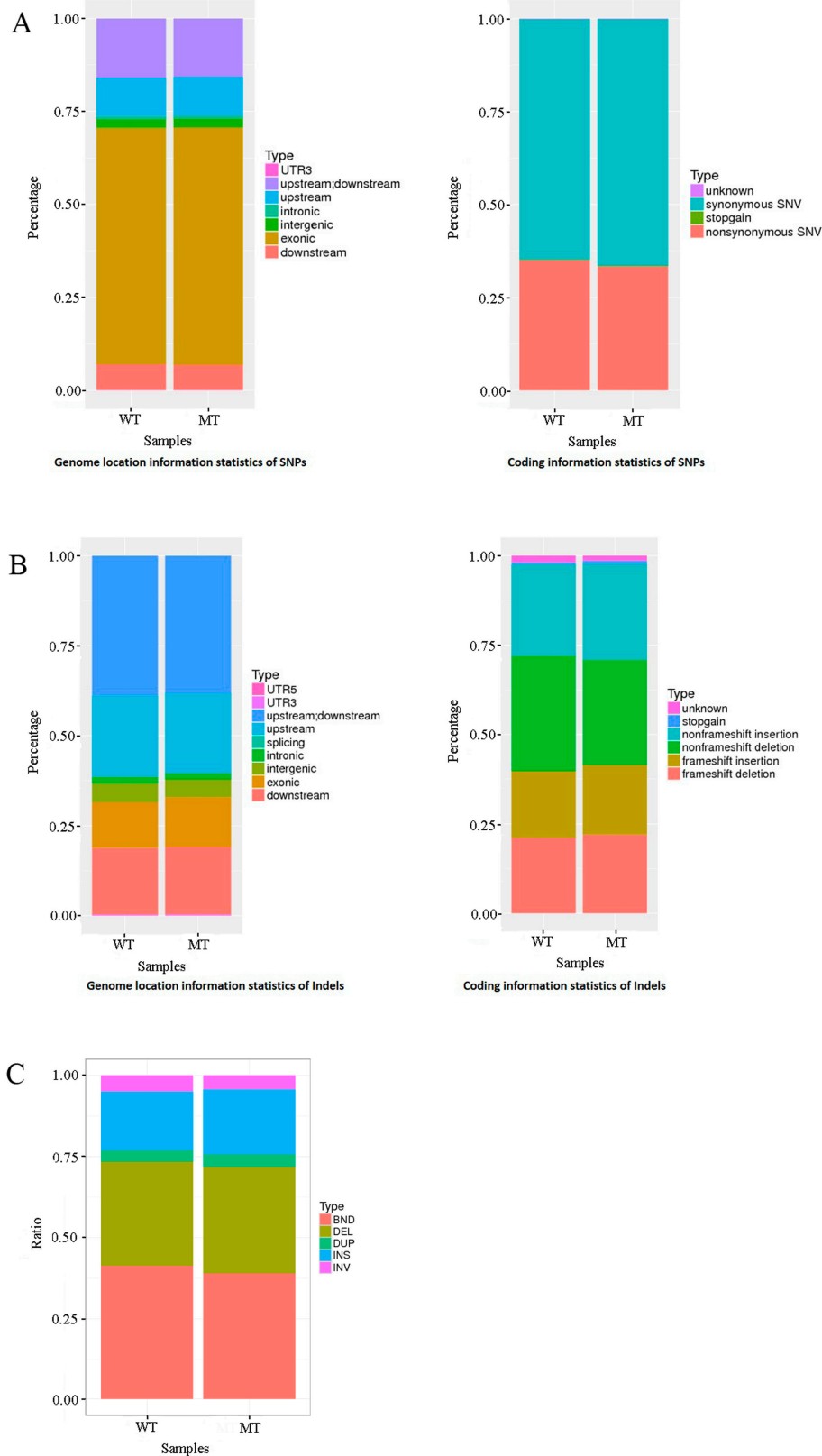

**Figure 2.** Genome sequencing results. Location and coding information of SNPs (**A**), location and coding information of InDels (**B**), and SV of the two strains (**C**). SNPs: Single Nucleotide Polymorphisms; InDels: Insertion-Deletion Mutations.

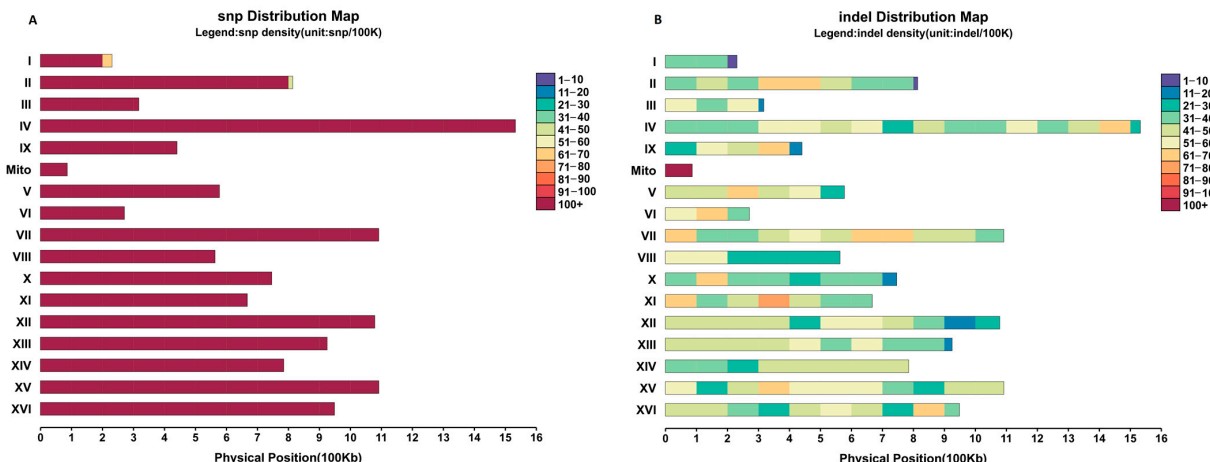

**Figure 3.** The distribution density of SNPs and InDels on chromosomes in MT strain. The distribution density of SNPs (**A**), the distribution density of InDels (**B**). SNP: Single Nucleotide Polymorphism; InDel: Insertion–Deletion Mutation.

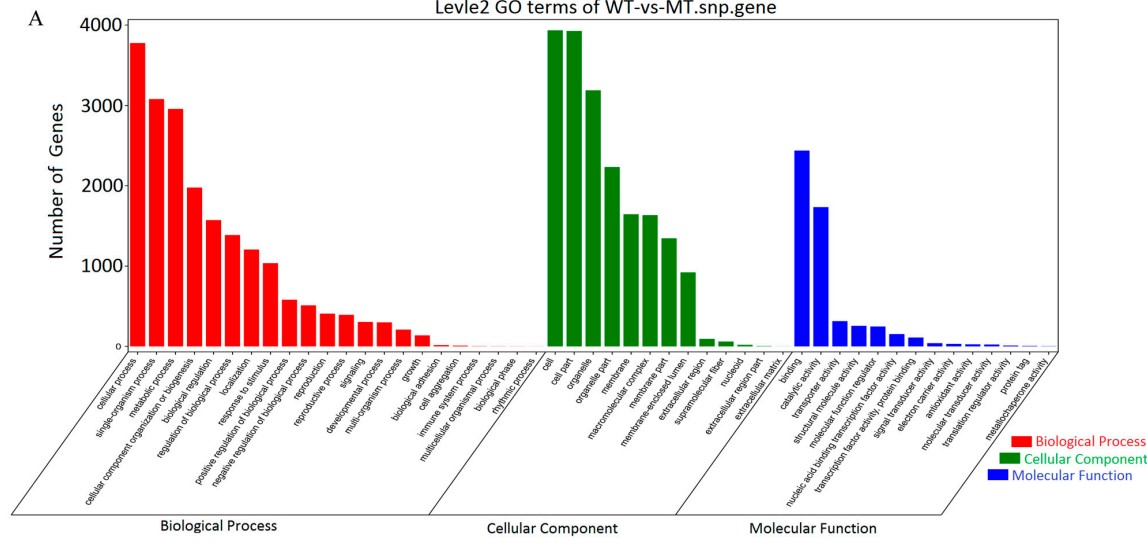

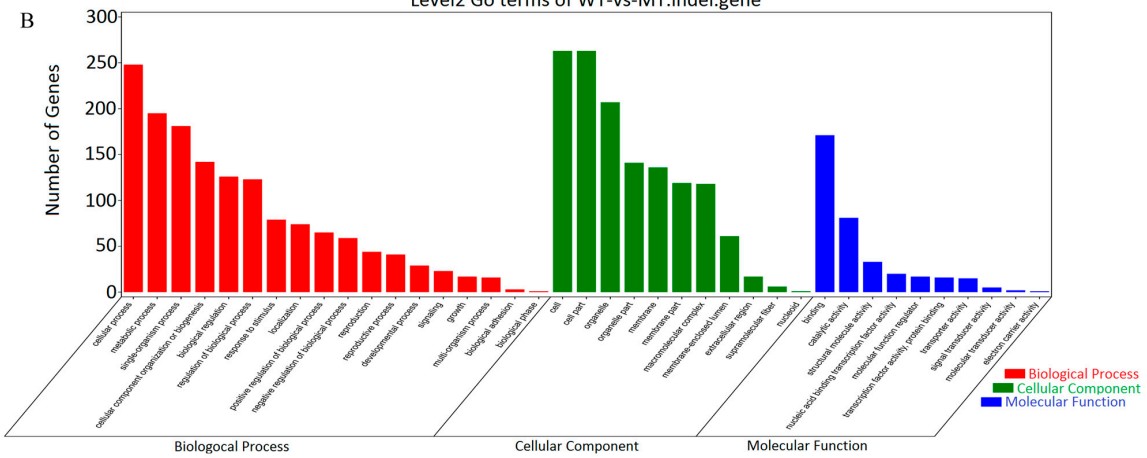

**Figure 4.** GO enrichment of the genes associated with SNPs and InDels in MT strain. GO enrichment of the genes associated with SNPs (**A**), GO enrichment of the genes associated with InDels (**B**). GO: Gene Ontology; SNP: Single Nucleotide Polymorphism; InDel: Insertion–Deletion Mutation.

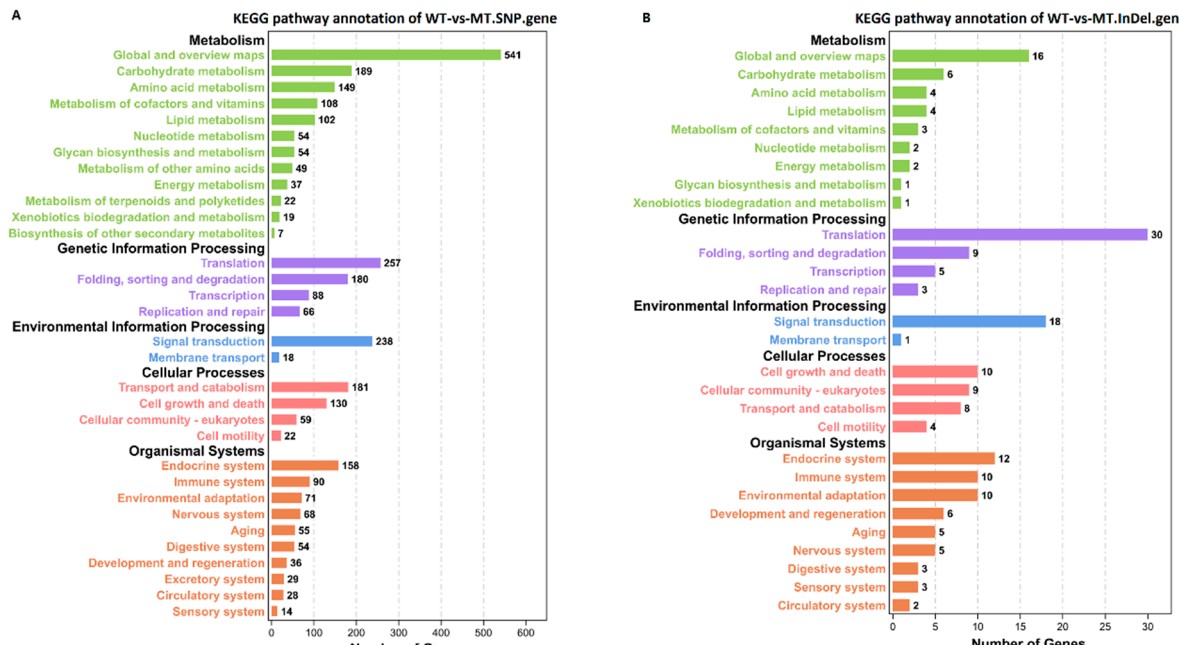

**Figure 5.** KEGG enrichment of the genes associated with SNPs and InDels in MT strain. KEGG enrichment of the genes associated with SNPs (**A**), KEGG enrichment of the genes associated with In-Dels (**B**). KEGG: Kyoto Encyclopedia of Genes and Genomes; SNP: Single Nucleotide Polymorphism; InDel: Insertion–Deletion Mutation.

### 3.3. DEGs and KEGG Enrichment Analysis between Different Groups

The statistics on quality of sequencing data were shown in Tables S11–S15. Plot FPKM distribution for all samples were shown in Figure S1. The heat map of the Pearson correlation coefficient R between sanmples was shown in Figure S2. Genes with a false discovery rate ≤ 0.05 and an absolute fold change ≥ 2 were considered DEGs. DEGs between different comparison groups are shown in Figure 6. Compared to the WT strain, the MT strain had significantly upregulated gene expression at the T1 time point and significantly downregulated gene expression at the T2 and T3 time points. There were more upregulated genes in the WT strain and more downregulated genes in the MT strain at the T2 and T3 time points than at the T1 time point. For both the WT and MT strains, there was little expression difference between the T2 and T3 time points.

The top 20 metabolism pathways of the enriched DEGs between the WT and MT strains at three time points are shown in Figure 7. The enriched genes were mainly involved in sugar, amino acid, and fatty acid metabolism. Between WT-T1 and MT-T1, the vitamin B6 metabolism was significantly enriched, and starch and sucrose metabolism, Purine metabolism, and glycolysis/gluconeogenesis were the most enriched pathways. The citrate cycle and pyruvate metabolism pathways were significantly enriched between WT-T2 and MT-T2; the citrate cycle, pyruvate metabolism, and glycolysis/gluconeogenesis pathways collectively enriched the most genes between the two strains. Between WT-T3 and MT-T3, the significantly enriched pathway was alanine, aspartate, and glutamate metabolism, together with pyruvate metabolism, citrate cycle, glycolysis/gluconeogenesis, cysteine, and methionine metabolism enriched the most genes between the two strains. The results were consistent with the enrichment of genomic variation genes, including SNPs and InDels.

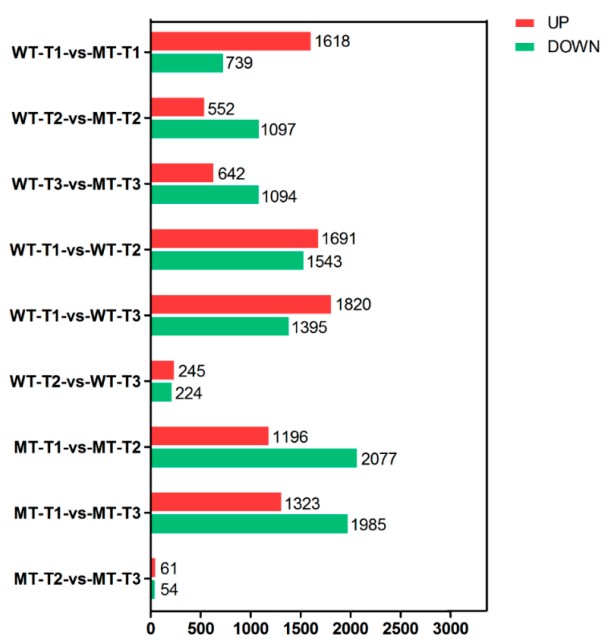

**Figure 6.** Statistics of the DEGs between different comparison groups. DEGs: different expression genes.

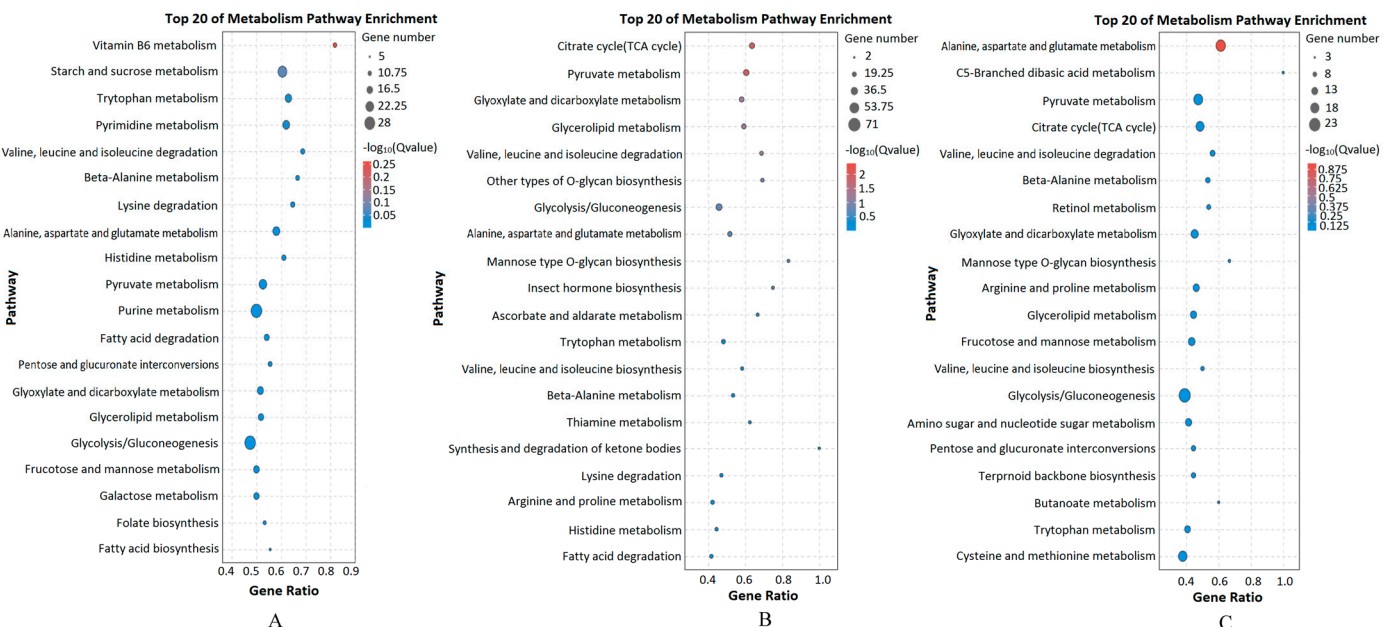

**Figure 7.** KEGG enrichment of the DEGs between WT and MT strains. WT-T1 vs. MT-T1 (**A**), WT-T2 vs. MT-T2 (**B**), and WT-T3 vs. MT-T3 (**C**). KEGG: Kyoto Encyclopedia of Genes and Genomes; DEGs: different expression genes.

### 3.4. Expression Trends of DEGs on the Timeline and Exclusive Genes in the MT Strain

To examine the expression patterns of DEGs, the expression data of each sample (in the order of treatment) were normalized to 0, log2(v1/v0), log2(v2/v0), and then clustered using STEM. The parameters of "Maximum unit change in model profiles between time points" = 1, "Maximum output profiles number" = 20, and "Minimum ratio of fold change of DEGs" was ≥2.0. The clustered profiles with *p*-values ≤ 0.05 were considered significant profiles. There were eight expression trend profiles, and four profiles were significantly enriched for each strain for the fermentation process: profile 0—downregulation; profile 1—initial downregulation and then no change; profile 6—initial upregulation and then no change; and profile 7—upregulation (Figure 8).

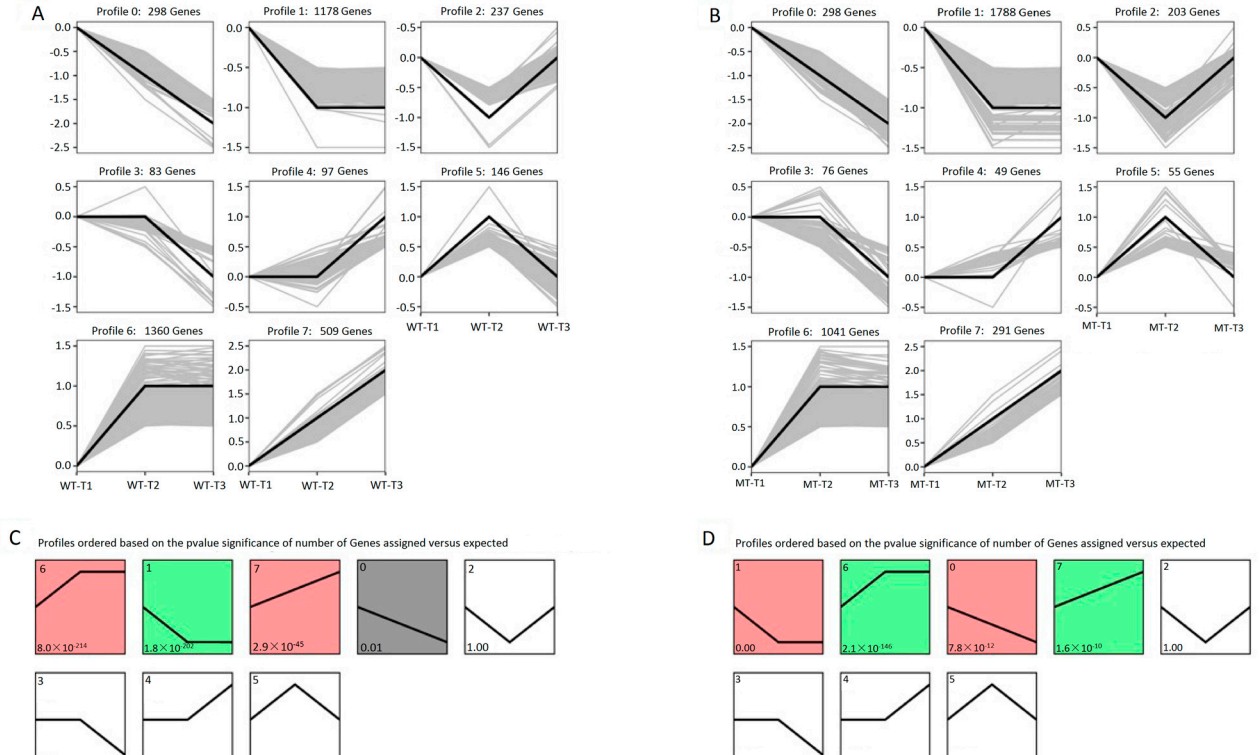

**Figure 8.** Gene expression trends of DEGs between the two strains during fermentation. The black line is the trend line, and the gray lines are gene lines. Gene expression trends of DEGs in the WT strain (**A**), profiles of the WT strain ordered by *p*-value significance of the number of genes assigned versus expected (**C**), gene expression trends of DEGs in the MT strain (**B**), profiles of the MT strain ordered by *p*-value significance of the number of genes assigned versus expected (**D**). DEGs: different expression genes.

Exclusive genes in the MT strain of each significantly enriched profile were determined using Venn diagram analysis, and KEGG pathway enrichment analysis of the exclusive genes was conducted. As shown in Figure 9, the exclusive genes in profiles 6 and 7 with upregulated trends are mainly enriched in carbon and sugar metabolism pathways, whereas the exclusive genes in profiles 0 and 1 with downregulated trends are mainly enriched in the ribosome and amino acid and fatty acid biosynthesis. Genes involved in the top 15 metabolic pathways are shown in Table 1.

The interaction network of the genes in Table 1 was analyzed, except for those involved in ribosomes. The Pearson correlation and significance between pairwise genes were calculated, and the top 100 pairs of absolute correlation were shown when the *p*-value was ≤0.05. As shown in Figure 10, there were nine groups in the interaction network. The largest group consisted of 45 genes and centered on *SPC*3 and *PGI*1, the two central blocks being connected by the protection of telomeres 1 (*POT*1) and GMP synthase (*GUA*1). The second group consisted of seven genes, with *PFK*2 as the hub gene. The third group consisted of six genes, with *GRE*3 as the hub gene. There were low numbers of genes and relatively simple relationships in the remaining six groups.

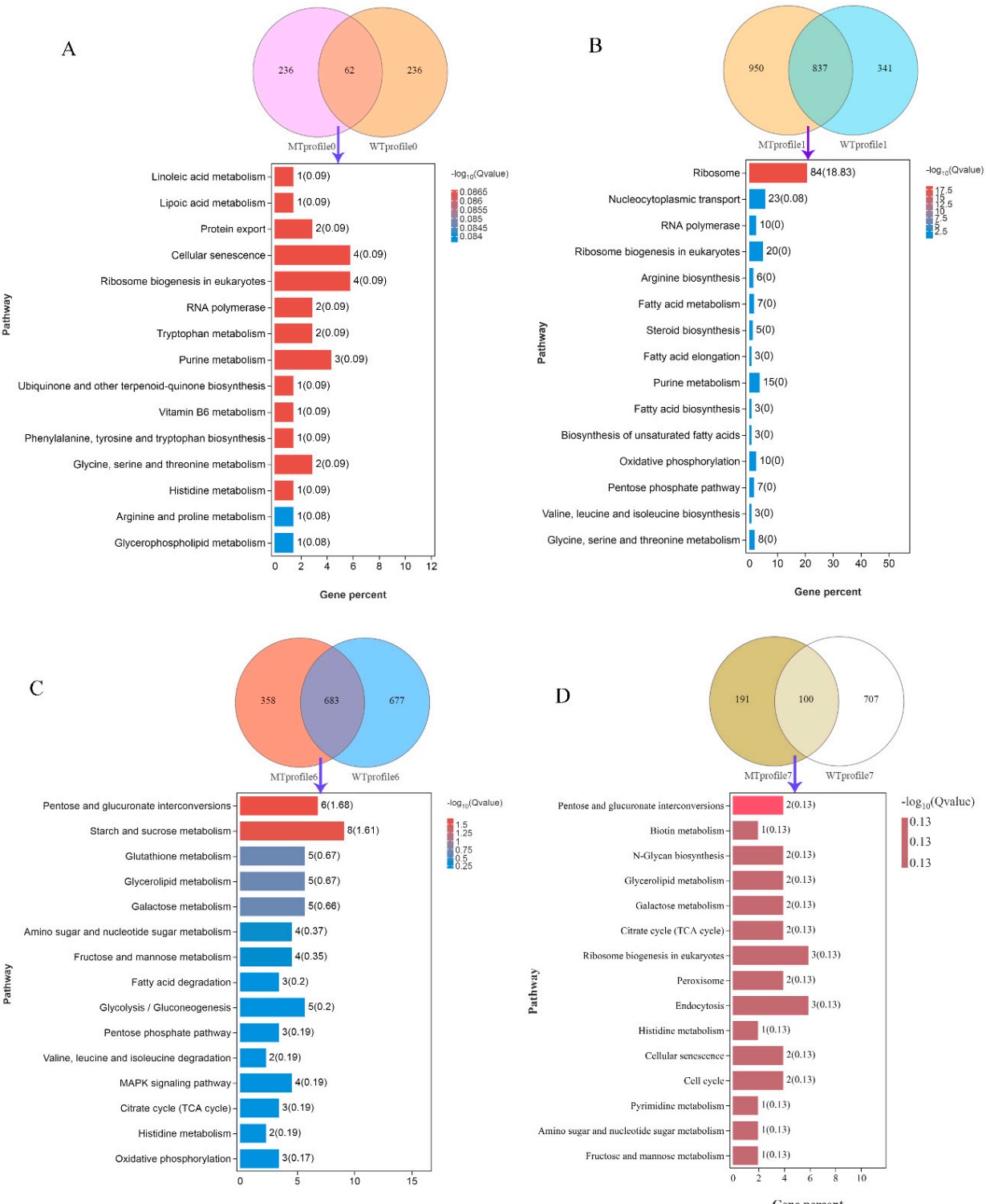

**Figure 9.** Exclusive genes by Venn diagrams and KEGG pathway enrichment of these exclusive genes in the MT strain. Venn diagram analysis of MTprofile0 and WT profile0, and the top 15 pathway enrichments of the exclusive genes in the MT strain (**A**), Venn diagram analysis of MTprofile1 and WT profile1, and the top 15 pathway enrichments of the exclusive genes in the MT strain (**B**), Venn diagram analysis of MTprofile6 and WT profile6, and the top 15 pathway enrichments of the exclusive genes in the MT strain (**C**), Venn diagram analysis of MTprofile7 and WT profile7, and the top 15 pathway enrichments of the exclusive genes in the MT strain (**D**). KEGG: Kyoto Encyclopedia of Genes and Genomes; DEGs: different expression genes.

**Table 1.** Genes involved in the top 15 enriched pathways of the exclusive genes in the MT strain.

| Pathway | Gene ID | Gene Symbol |
| --- | --- | --- |
| | Profile 0 [a] | |
| Linoleic acid metabolism | YOL011W | PLB3 |
| Lipoic acid metabolism | YLR239C | LIP2 |
| Protein export | YMR150C, YLR066W | IMP1, SPC3 |
| Cellular senescence | YPR119W, YBR085, WYKL168C, YBR195C | CLB2, AAC3, KKQ8, MSI1 |
| Ribosome biogenesis in eukaryotes | YLR129W, YLR175W, YDL014W, YJL069C | DIP2, CBF5, NOP1, UTP18 |
| RNA polymerase | YNL248C, YGL070C | RPA49, RPB9 |
| Tryptophan metabolism | YLR044C, YGR088W | PDC1, CTT1 |
| Purine metabolism | YEL042W, YIR029W, YOR128C | GDA1, DAL2, ADE2 |
| Ubiquinone and other terpenoid-quinone biosynthesis | YDR539W | FDC1 |
| Vitamin B6 metabolism | YNL334C | SNO2 |
| Phenylalanine, tyrosine, and tryptophan biosynthesis | YDR007W | TRP1 |
| Glycine, serine, and threonine metabolism | YML082W, YMR189W | YML082W, GCV2 |
| Histidine metabolism | YCL030C | HIS4 |
| Arginine and proline metabolism | YLR142W | PUT1 |
| Glycerophospholipid metabolism | YOL011W | PLB3 |
| | Profile 1 [b] | |
| Ribosome | YGR085C, YDR418W, YIL133C, YNL069C, YDL082W, YHL001W, YJL177W, YMR242C, YOR312C, YNL301C, YBL027W, YBR191W, YFL034C-A, YOL127W, YBL087C, YER117W, YGL031C, YLR009W, YGR034W, YLR344W, YDR471W, YHR010W, YGL030W, YDL075W, YLR406C, YBL092W, YER056C-A, YIL052C, YOR234C, YPL143W, YDL191W, YDL136W, YMR194W, YPR043W, YLR185W, YDR500C, YLR325C, YKR094W, YIL148W, YNL162W, YHR141C, YML073C, YLR448C, YPL198W, YFR031C-A, YIL018W, YGL147C, YLR340W, YDL081C, YOL039W, YOR293W, YDR025W, YCR031C, YDR337W, YLR367W, YOL040C, YDL083C, YMR143W, YDR447C, YML024W, YML026C, YDR450W, YNL302C, YOL121C, YJL136C, YKR057W, YPR132W, YER074W, YLR333C, YGR027C, YHR021C, YLR264W, YOR167C, YDL061C, YOR182C, YNL178W, YHR203C, YPL090C, YBR181C, YNL096C, YBL072C, YBR189W, YGR214W, YLR048W | RPL11B, RPL12B, RPL16A, RPL16B, RPL13A, RPL14B, RPL17B, RPL20A, RPL20B, RPL18B, RPL19B, RPL21A, RPL22B, RPL25, RPL23A, RPL23B, RPL24A, RLP24, RPL26B, RPL26A, RPL27B, RPL27A, RPL30, RPL31A, RPL31B, RPL32, RPL34A, RPL34B, RPL33B, RPL33A, RPL35A, RPL35B, RPL36A, RPL43A, RPL37A, RPL37B, RPL38, RPL40B, RPL40A, RPL42A, RPL42B, RPL6A, RPL6B, RPL7B, RPL2A, RPL2B, RPL9A, RPP0, RPP1A, RPP2A, RPS10A, RPS11A, RPS14A, MRPS28, RPS22B, RPS15, RPS16B, RPS16A, RPS17B, RPS17A, RPS18B, RPS18A, RPS19B, RPS19A, RPS21B, RPS21A, RPS23B, RPS24A, RPS25B, RPS25A, RPS27B, RPS28B, RPS28A, RPS29B, RPS30B, RPS3, RPS4B, RPS6A, RPS6B, RPS7B, RPS8A, RPS9B, RPS0A, RPS0B |
| Nucleocytoplasmic transport | YER168C, YNL221C, YNL244C, YMR260C, YJR007W, YOR260W, YER025W, YMR146C, TDR429C, YMR309C, YOR361C, YBR079C, YJL138C, YOL139C, YGR162W, YPR041W, YGR083C, YLR208W, YKL205W, YGL092W, YKL068W, YFR002W, YMR235C | CCA1, POP1, SUI1, TIF11, SUI2, GCD1, GCD11, TIF34, TIF35, NIP1, PRT1, RPG1, TIF2, CDC33, TIF4631, TIF5, GCD2, SEC13, LOS1, NUP145, NUP100, NIC96, RNA1 |

**Table 1.** *Cont.*

| Pathway | Gene ID | Gene Symbol |
|---|---|---|
| RNA polymerase | YOR341W, YJR063W, YPR010C, YIL021W, YPR187W, YOR224C, YNL113W, YNR003C, YDL150W, YPR110C | RPA190, RPA12, RPA135, RPB3, RPO26, RPB8, RPC19, RPC34, RPC53, RPC40 |
| Ribosome biogenesis in eukaryotes | YNL221C, YIL035C, YHR089C, YDL208W, YLR059C, YNL132W, YNL163C, YGR090W, YCL031C, YDR398W, YDR324C, YMR093W, YLR222C, YCR057C, YLR197W, YDR339C, YML093W, YLR186W, YLR022C, YLR397C | POP1, CKA1, GAR1, NHP2, REX2, KRE33, RIA1, UTP22, RRP7, UTP5, UTP4, UTP15, UTP13, PWP2, NOP56, FCF1, UTP14, EMG1, SDO1, AFG2 |
| Arginine biosynthesis | YOR375C, YDR111C, YPR035W, YOL058W, YER069W, YKL106W | ARG5,6, ALT2, GDH1, GLN1, ARG1, AAT1 |
| Fatty acid metabolism | YIL009W, YMR246W, YAR035W, YER061C, YLR372W, YJL196C, YJL097W | FAA3, FAA4, YAT1, CEM1, YLR372W, ELO1, PHS1 |
| Steroid biosynthesis | YLR056W, YLR020C, YML008C, YLL012W, YGR060W | ERG3, YEH2, ERG6, YEH1, ERG25 |
| Fatty acid elongation | YLR372W, YJL196C, YJL097W | YLR372W, ELO1, PHS1 |
| Purine metabolism | YJR105W, YJR069C, YBL068W, YMR120C, YER070W, YMR300C, YJL070C, YML022W, YMR217W, YER005W, AR015W, YGL234W, YHL011C, YCR026C, YNL141W | ADO1, HAM1, PRS4, ADE17, RNR1, ADE4, YMR120C, APT1, GUA1, YND1, ADE1, ADE5,7, PRS3, NPP1, AAH1 |
| Fatty acid biosynthesis | YIL009W, YMR246W, YER061C | FAA3, FAA4, CEM1 |
| Biosynthesis of unsaturated fatty acids | YLR372W, YJL196C, YJL097W | YLR372W, ELO1, PHS1 |
| Oxidative phosphorylation | YOR270C, YBR039W, YDL004W, YGL008C, YKL141W, YGR020C, YLR447C, YDL185W, YBL099W, YDR529C | VPH1, ATP3, ATP16, PMA1, SDH3, VMA7, VMA1, ATP1, QCR7 |
| Pentose phosphate pathway | YBL068W, YMR205C, YCR073W-A, YGR240C, YGL185C, YBR196C, YHL011C | PRS4, PFK2, SOL2, PFK1, YGL185C, PGI1, PRS3 |
| Valine, leucine, and isoleucine biosynthesis | YNL104C, YOR108W, YLR355C | LEU4, LEU9, ILV5 |
| Glycine, serine, and threonine metabolism | YDR232W, YDR158W, YOL056W, YCR053W, YOR184W, YGL185C, YGR155W, YAL044C | HEM1, HOM2, GPM3, THR4, SER1, YGL185C, CYS4, GCV3 |
| | Profile 6 [c] | |
| Pentose and glucuronate interconversions | YGR194C, YKL035W, YJR096W, YAL061W, YHL012W, YHR104W | XKS1, UGP1, YJR096W, BDH2, YHL012W, GRE3 |
| Starch and sucrose metabolism | YDL243C, YCR107W, YKL035W, YEL011W, YDR001C, YFR053C, YHL012W, YLR258W | AAD4, AAD3, UGP1, GLC3, NTH1, HXK1, YHL012W, GSY2 |
| Glutathione metabolism | YER163C, YGR180C, YLR174W, YIR038C, YKL026C | GCG1, RNR4, IDP2, GTT1, GPX1 |
| Glycerolipid metabolism | YDR018C, YKL094W, YPL061W, YJR096W, YHR104W | YDR018C, YJU3, ALD6, YJR096W, GRE3 |
| Galactose metabolism | YKL035W, YJR096W, YFR053C, YHL012W, YHR104W | UGP1, YJR096W, HXK1, YHL012W, GRE3 |
| Amino sugar and nucleotide sugar metabolism | YKL035W, YMR084W, YFR053C, YHL012W | UGP1, YMR084W, HXK1, YHL012W |
| Fructose and mannose metabolism | YJR096W, YFR053C, YAL061W, YHR104W | YJR096W, HXK1, BDH2, GRE3 |
| Fatty acid degradation | YPL061W, YDL168W, YIL160C | ALD6, SFA1, POT1 |
| Glycolysis/Gluconeogenesis | YDL243C, YCR107W, YPL061W, YDL168W, YFR053C | AAD4, AAD3, ALD6, SFA1, HXK1 |
| Pentose phosphate pathway | YGR248W, YBR117C, YGR043C | SOL4, TKL2, NQM1 |
| Valine, leucine and isoleucine degradation | YPL061W, YIL160C | ALD6, POT1 |

**Table 1.** *Cont.*

| Pathway | Gene ID | Gene Symbol |
| --- | --- | --- |
| MAPK signaling pathway | YDL006W, YNL098C, YBL016W, YLL024C | PTC1, RAS2, FUS3, SSA2 |
| Citrate cycle (TCA cycle) | YLR164W, YLR174W, YJL045W | SHH4, IDP2, YJL045W |
| Histidine metabolism | YPL061W, YNL092W | ALD6, YNL092W |
| Oxidative phosphorylation | YLR164W, Q0130, YJL045W | SHH4, OLI1, YJL045W |
| | Profile 7 [d] | |
| Pentose and glucuronate interconversions | YOR120W, YNR073C | GCY1, MAN2 |
| Biotin metabolism | YNR058W | BIO3 |
| N-Glycan biosynthesis | YBR110W, YBR070C | ALG1, ALG14 |
| Glycerolipid metabolism | YOR120W, YOR245C | GCY1, DGA1 |
| Galactose metabolism | YOR120W, YNR071C | GCY1, YNR071C |
| Citrate cycle (TCA cycle) | YCR005C, YNL009W | CIT2, IDP3 |
| Ribosome biogenesis in eukaryotes | YHR170W, YNL075W, YLR106C | NMD3, IMP4, MDN1 |
| Peroxisome | YNL009W, YJR104C | IDP3, SOD1 |
| Endocytosis | YHL002W, YOR211C, YER125W | HSE1, MGM1, RSP5 |
| Histidine metabolism | YMR209C | YMR209C |
| Cellular senescence | YCR008W, YJR066W | SAT4, TOR1 |
| Cell cycle | YJL187C, YCR008W | SWE1, SAT4 |
| Pyrimidine metabolism | YDR020C | DAS2 |
| Amino sugar and nucleotide sugar metabolism | YBR023C | CHS3 |
| Fructose and mannose metabolism | YOR120W | GCY1 |

[a] Significant expression trend of the MT strain, profile 0-downregulation. [b] Significant expression trend of the MT strain, profile 1-initial downregulation, and then no change. [c] Significant expression trend of the MT strain, profile 6-initial upregulation, and then no change. [d] Significant expression trend of the MT strain, profile 7-upregulation.

As a hub gene in the first group, *SPC3* is involved in protein export. The connected genes complement factor B (*CFB5*) and small subunit processome component (*UTP18*) are involved in ribosome biogenesis in eukaryotes; *FAA3* is involved in fatty acid metabolism; lipase (*LIP2*) is involved in lipoic acid metabolism; *LEU9* is involved in valine, leucine, and isoleucine biosynthesis; phosphoribosylaminoimidazole carboxylase (*ADE2*) is involved in purine metabolism; and *VMA7* is involved in oxidative phosphorylation. These connections demonstrate the relationship among protein export, ribosome biogenesis, and oxidative phosphorylation, as well as fatty acid and amino acid metabolism. *SPC3* encodes a signal-anchored protein subunit that enters the endoplasmic reticulum (ER) as a signal peptidase [32]. The ER plays an important role in maintaining the balance and stability of cellular proteins; misfolded proteins accumulate in the ER when cells are under different kinds of stress, known as ER stress [33]. The role that *SPC3* and other interacting genes play in ER stress is worth investigating.

*PGI1* is involved in the pentose phosphate pathway, whereas the genes bifunctional purine biosynthetic protein (*ADE5,7*), bifunctional glutathione transferase (*GTT1*), 5-aminolevulinate synthase (*HEM1*), O-phosphol-L-serine:2-oxoglutarate transaminase (*SER1*), *SFA1*, *ELO1*, *CEM1*, and sterol 24-C-methyltransferase (*ERG6*) are involved in purine metabolism; glutathione metabolism; glycine, leucine, and isoleucine biosynthesis; and fatty acid metabolism. Genes involved in fatty acid biosynthesis (*ELO1* and *CEM1*) were downregulated, whereas *SFA1* involved in fatty acid degradation was upregulated, which is beneficial to the accumulation of acetyl-CoA. *SFA1* is also a bifunctional alcohol dehydrogenase, and its synergic upregulation with *ALD6* and *POT1* increases the catalytic degradation of fatty acids to alcohol. The effect of *SFA1* on ethanol production in yeast cells has been studied, but the mechanisms regulating the different effects under different conditions remain unclear [34]. This module mainly showed the relationship between the pentose phosphate pathway and fatty acid and amino acid metabolism. The Hsp70 family chaperone (*SSA2*) and *PTC1*, involved in the MAPK pathway, were located at the edge of this module. Glutathione transferase (*Gtt1*) of *S. cerevisiae* is crucial to the response to

hydrogen peroxide stress. Increasing glutathione content might enhance yeast cell tolerance to lignocellulose inhibitors and increase the production of ethanol [35,36]. *GTT1* expression was negatively correlated with interlinked genes in the network of this module; these interactions help to better understand *GTT1*.

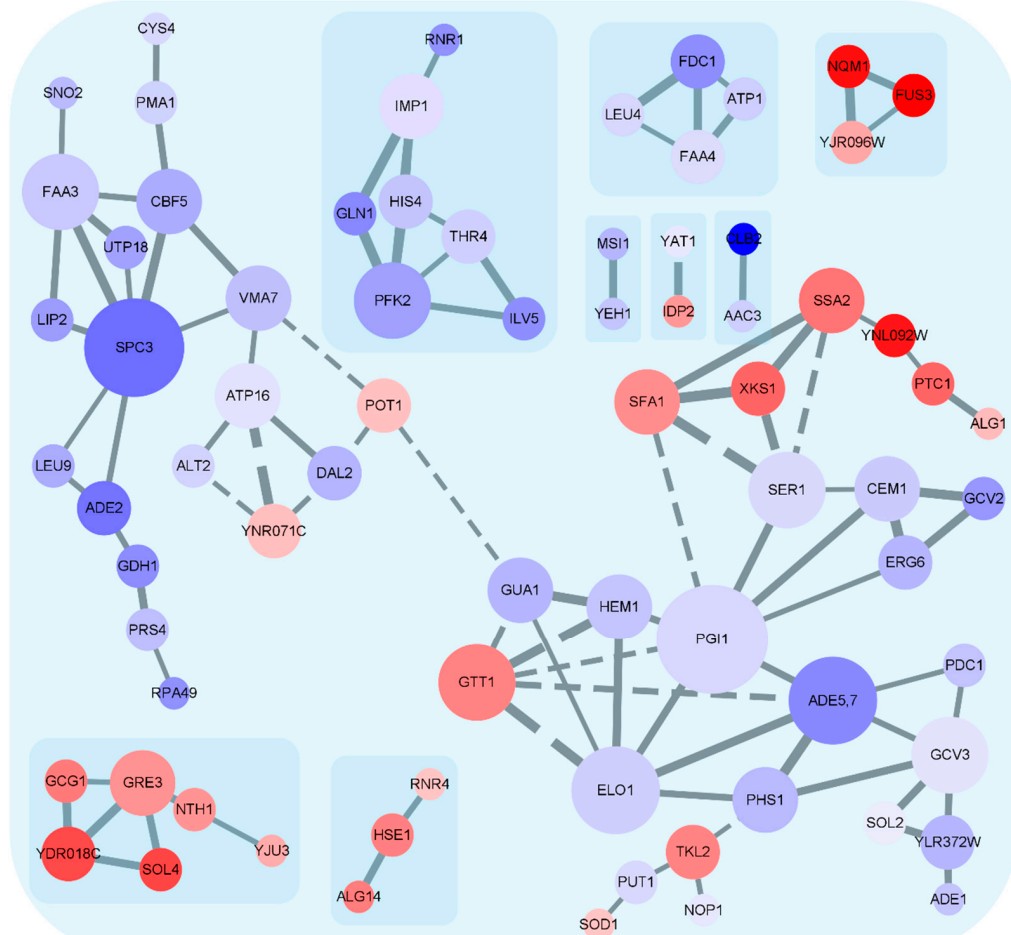

**Figure 10.** Protein–protein interaction network of the exclusive genes in the profile 0, 1, 6, and 7 of the MT strain. The node circles represent genes, labeling with gene ID or gene symbol; the node size is graded according to the Cytoscape connectivity, with greater connectivity leading to larger nodes. The node color is gradually changed according to log2FC; red represents log2FC > 0; the darker the red, the larger the upregulation ratio; blue represents log2FC < 0; the bluer the color, the larger the downregulation ratio. Positive correlations are shown by solid gray lines, and negative correlations are shown by dashed gray lines. The thickness of the line gradually changes according to the absolute value of the correlation coefficient; the larger value, the thicker line.

The second group comprised the hub gene *PFK2*, with glutamate–ammonia ligase (*GLN1*), trifunctional histidinol dehydrogenase (HIS4), threonine synthase (*THR4*), *ILV5*, endopeptidase catalytic subunit (*IMP1*), and ribonucleotide–diphosphate reductase subunit (*RNR1*). This group demonstrated the relationship between the pentose phosphate pathway (*PFK2*), amino acid metabolism (*GLN1*, *HIS4*, *THR4*, *ILV5*, *RNR1*), and protein export (*IMP1*).

The third group comprised *GRE3*, gamma–glutamylcyclotransferase (*GCG1*), putative acyltransferase (*YDR018C*), 6-phosphogluconolactonase (*SOL4*), *NTH1*, and acylglycerol lipase (*YJU3*), and demonstrated the connection between pentose and hexose metabolism. *GRE3* was the hub gene and was directly linked to four of these genes. An engineered *S. cerevisiae* strain overexpressing *GRE3*, xylitol dehydrogenase (*XYL2*), and xylulokinase

(*XYL3*) ferments xylose resulting in high ethanol yields [37]. Thus, *GRE3* has a coordinated role in ethanol fermentation in yeast using different sugars.

*PGI1*, *SPC3*, *PFK2,* and *GRE3* were the major hub genes in the network. In the future, these metabolic pathways may be optimized by overexpressing or deleting these hub genes to obtain *S. cerevisiae* strains with a higher capacity for ethanol fermentation.

### 3.5. qRT-PCR

qRT-PCR of 21 genes from two comparison groups was carried out to verify the accuracy of the RNA-Seq data and the expression level of important genes. The log2 n value of each gene is shown in Figure 11. All gene expression determined using qPCR was consistent with that obtained from RNA-Seq, thereby indicating the credibility of the RNA-Seq data. In the MT strain, the expression of sucrose invertase gene *SUC2* was downregulated; furthermore, the content of total sugar and reducing sugar in MT fermentation broth was lower than that of the WT strain after 16 h. The difference in sucrose utilization between the two strains highlights the need for further study.

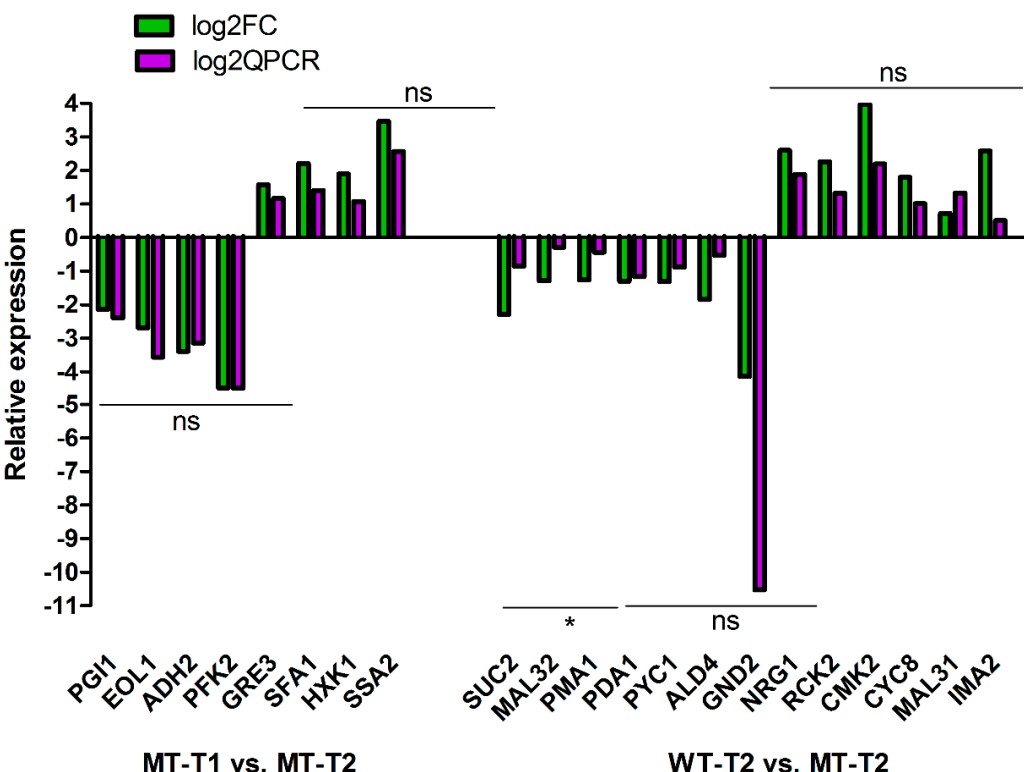

**Figure 11.** Consistency analysis of RNA-Seq and qPCR. (log2QPCR compared to log2FC, ns: not significant, * $p > 0.05$).

## 4. Discussion

### 4.1. Sucrose Consumption of the WT and MT Strains

*S. cerevisiae* consumes sucrose in two ways. (1) The hydrolyzation of sucrose into glucose and fructose by extracellular sucrose invertase encoded by the *SUC* gene family. In this process, monosaccharides enter cells by facilitated diffusion to participate in the carbon metabolism pathway; (2) Sucrose enters cells via the proton-coupled transporter and is hydrolyzed in the cytosol by maltose metabolizing enzyme and intracellular sucrose invertase [5,38]. The sucrose invertase gene *SUC2*, maltose permease genes (*MALx1*), maltase genes (*MALx2*), and some regulatory genes (e.g., *MALx3*) are the key genes in sucrose consumption of *S. cerevisiae* [38]. Invertases of *S. cerevisiae* coded by *SUC2* can be transcribed into two mRNAs that differ in their 5′ ends (1.8 Kb and 1.9 Kb). The longer of the two invertases with a signal peptide can be secreted outside the cell, and its synthesis level is

regulated by glucose repression. The shorter invertase lacking a signal peptide encodes the intracellular invertase [39,40]. In the absence of extracellular invertase activity of *S. cerevisia*, sucrose was internalized by proton symporters using ATP, which led to improving the anaerobic fermentation and ethanol yield from sugar [5]. In addition, the replacement of all endogenous hexose transporters with hexose–proton symport and extracellular invertase (*SUC2*) can increase ethanol yield and anaerobic growth of *S. cerevisiae* [41].

In this study, both RNA sequencing and qPCR showed that the MT strain had lower expression of *SUC2*, *PMA1* (H (+)-exporting P2-type ATPase), *MAL32* genes, and higher expression of *MAL31*, *IMA2* (oligo-1,6-glucosidase) genes than that of the WT strain during fermentation. Wesley Leoricy Marques et al. constructed several mutants to investigate sucrose metabolism in *S. cerevisiae*. Mutants IMU051 (*malΔ mphΔ suc2Δ*), IMU054 (*malΔ mphΔ suc2Δ MAL12*), and IMK700 (*malΔ mphΔ suc2Δ MAL11 imaΔ cas9*) could not grow on mediums with sucrose as the sole carbon source; however, the growth of mutants IMU055 (*malΔ mphΔ suc2Δ MAL11 MAL12*) and IMU048 (*malΔ mphΔ suc2Δ MAL11*) was not limited by these types of medium [38]. These results show the importance of maltose permease (*MALx1*) and oligo-1,6-glucosidase (*IMAx*) for sucrose consumption in *S. cerevisiae*. However, the transportation of sucrose by *MAL31* and hydrolyzation by *IMA2* in the MT strain requires verification.

We analyzed the protein interaction network of DEGs (WT-T2 vs. MT-T2) and extracted the network of genes related to sucrose consumption in Table 2. Pearson correlations between pairwise genes of DEGs were calculated. Considering the genes in Table 2 as core genes, the top 10 correlation pairs in the absolute value of each core gene were shown when the *p*-value was $\leq 0.05$. As shown in Figure 12, most core genes were connected by complex nodes. *MAL33*, *MPH2*, and *IMA2* showed significant upregulation. PMP2, *PGM1*, *HXT11*, *HRK1*, *MAL31*, *GRE3*, and *HXK2* showed marginal upregulation. The remaining core genes were downregulated. Among the core genes, *RGT2*, *AST2*, *HXT2*, and *HXT5* are closely related to the *SUC2* gene. All of them are related to the plasma membrane and glucose transporter. As a glucose-sensing receptor, the glucose-sensing signal generated by *RGT2* can lead to the inhabitation of the *RGT1* transcriptional repressor and, thus, the derepression of *HXT* genes that encode glucose transporters [42,43]. *RGT2* was downregulated in response to glucose starvation [42]. Does the down expression of *SUC2*, *RGT2*, and *HXTx* indicate that the sucrose is primarily hydrolyzed intracellularly in the MT strain? However, plasma membrane H+-ATPase *PMA1* and *PMA2*, which are related to the transmembrane transport of sucrose, are also downregulated in the MT strain. How the MT strain efficiently uses sucrose to produce ethanol is our next research objective. Our network diagram results revealed the potential pathway of sucrose absorption and provided a basis for subsequent study of the sucrose mode of absorption.

**Table 2.** Genes of DEGs between WT-T2 vs. MT-T2 are possibly related to sucrose consumption.

| Gene Name | Description | FDR | Log2-Fold Change [a] |
|---|---|---|---|
| SUC2 | Invertase; sucrose hydrolyzing enzyme | $1.08 \times 10^{-20}$ | −2.28 |
| MAL11 | High-affinity maltose transporter (alpha-glucoside transporter) | 0.205 | −0.91 |
| MAL31 | Maltose permease; high-affinity maltose transporter (alpha-glucoside transporter) | $6.65 \times 10^{-5}$ | 0.72 |
| MAL12 | Maltase (alpha-D-glucosidase) | 0.141 | −4.03 |
| MAL32 | Maltase (alpha-D-glucosidase) | 0.466 | −1.29 |
| MAL33 | MAL-activator protein | $5.67 \times 10^{-28}$ | 2.87 |
| MAL13 | MAL-activator protein | $6.15 \times 10^{-4}$ | 0.55 |

**Table 2.** *Cont.*

| Gene Name | Description | FDR | Log2-Fold Change [a] |
|---|---|---|---|
| PMA1 | Plasma membrane P2-type H+-ATPase | $1.98 \times 10^{-4}$ | −1.26 |
| PMA2 | Plasma membrane H+-ATPase | 0.925 | −0.53 |
| PMP2 | Proteolipid associated with plasma membrane H(+)-ATPase (Pma1p) | 0.379 | 1.39 |
| SOP4 | ER-membrane protein | 0.935 | −0.46 |
| AST1 | Lipid raft-associated protein; interacts with the plasma membrane ATPase Pma1p and has a role in its targeting of the plasma membrane by influencing its incorporation into lipid rafts | $2.32 \times 10^{-27}$ | −5.37 |
| AST2 | Lipid raft-associated protein; overexpression restores Pma1p localization to lipid rafts which are required for targeting Pma1p to the plasma membrane | $1.54 \times 10^{-5}$ | −2.36 |
| HRK1 | Protein kinase; implicated in activation of the plasma membrane H(+)-ATPase Pma1p in response to glucose metabolism | $1.4 \times 10^{-8}$ | 1.00 |
| IMA1 | Major isomaltase (alpha-1,6-glucosidase/alpha-methylglucosidase) | 0.186 | −1.25 |
| IMA2 | Isomaltase (alpha-1,6-glucosidase/alpha-methylglucosidase) | $1.28 \times 10^{-34}$ | 2.58 |
| IMA3 | Alpha-glucosidase; weak but broad substrate specificity for alpha-1,4- and alpha-1,6-glucosides | 0.697 | −0.27 |
| IMA4 | Alpha-glucosidase; weak but broad substrate specificity for alpha-1,4- and alpha-1,6-glucosides | 0.697 | −0.27 |
| IMA5 | Alpha-glucosidase; specificity for isomaltose, maltose, and palatinose, but not alpha-methylglucoside | 0.813 | −0.21 |
| MPH2 | Alpha-glucoside permease | 1 | 3.32 |
| MPH3 | Alpha-glucoside permease | 0.397 | 0.087 |
| PFK2 | Beta subunit of heterooctameric phosphofructokinase | $6.89 \times 10^{-22}$ | −2.41 |
| GRE3 | Aldose reductase; involved in methylglyoxal, d-xylose, arabinose, and galactose metabolism | $1.45 \times 10^{-9}$ | 0.62 |
| PGM1 | Phosphoglucomutase, minor isoform; catalyzes the conversion from glucose-1-phosphate to glucose-6-phosphate | $3.2 \times 10^{-8}$ | 1.16 |

**Table 2.** *Cont.*

| Gene Name | Description | FDR | Log2-Fold Change [a] |
|---|---|---|---|
| GAL2 | Galactose permease | 0.017 | −2.12 |
| RGT2 | Plasma membrane high glucose sensor that regulates glucose transport | $5.51 \times 10^{-11}$ | −2.54 |
| HXT1 | Low-affinity glucose transporter of the major facilitator superfamily | $1.08 \times 10^{-5}$ | −1.53 |
| HXT2 | High-affinity glucose transporter of the major facilitator superfamily | $2.19 \times 10^{-7}$ | −2.84 |
| HXT3 | Low-affinity glucose transporter of the major facilitator superfamily | 0.018 | −1.08 |
| HXT4 | High-affinity glucose transporter; member of the major facilitator superfamily | $2.79 \times 10^{-4}$ | −1.57 |
| HXT5 | Hexose transporter with moderate affinity for glucose | $3.54 \times 10^{-15}$ | −2.31 |
| HXT6 | High-affinity glucose transporter; member of the major facilitator superfamily | $1.99 \times 10^{-26}$ | −4.56 |
| HXT7 | High-affinity glucose transporter; member of the major facilitator superfamily | 0.414 | −0.83 |
| HXT11 | Hexose transporter | $2.96 \times 10^{-11}$ | 1.29 |
| HXK1 | Hexokinase isoenzyme 1 | 0.228 | −0.77 |
| HXK2 | Hexokinase isoenzyme 2 | $1.30 \times 10^{-2}$ | 0.85 |

[a] log 2-fold change of differential expression; a positive number means up expression, and a minus means down expression.

### 4.2. Carbon Metabolism in the MT Strain

The metabolic pathway enrichment results of exclusive genes in the MT strain were mainly reflected in amino acid, fatty acid, and sugar metabolism. The carbon skeleton of amino acids is broken down to form acetyl-CoA, α-ketoglutaric acid, succinyl-CoA, fumaric acid, and oxaloacetic acid, all of which enter the citric acid cycle. The amino acid synthesis also uses pyruvate, α-ketoglutaric acid, and oxaloacetic acid, which are intermediates in glycolysis, the citric acid cycle, and the pentose phosphate pathway, respectively. These intermediates link sugar metabolism to amino acid metabolism [44,45]. As a carbon skeleton, acetyl-CoA is the catabolic product of fatty acids and is the only source of carbon atoms in the fatty acid molecule. As shown in Figure 13, the expression of exclusive genes in the MT strain involved in amino acid, fatty acid, and sugar metabolism resulted in carbon metabolism flow to pyruvate and acetyl-CoA. Downregulation of ketol-acid reductoisomerase (*ILV5*), 2-isopropylmalate synthase (*LEU4*), 2-isopropylmalate synthase (*LEU9*), and dihydrolipoyllysine-residue acetyltransferase (*LAT1*) genes can reduce the synthesis of pyruvate to valine, leucine, isoleucine, and acetyl-CoA, respectively. Upregulation of xylulokinase (*XKS1*), 3-hydroxybutyrate dehydrogenase 2 (*BDH2*), aldo-keto reductase superfamily protein (*YJR096W*), and *GRE3* genes contribute to the conversion of ribose-5-phosphate to D-xylose, and subsequently to pyruvate. Pyruvate accumulation is beneficial to increase ethanol production. Upregulation of *POT*1 and downregulation of fatty acid synthase (*CEM1*), long-chain fatty acid-CoA ligases (*FAA3* and *FAA4*), fatty acid elongase (*ELO1*), carnitine O-acetyltransferase (*YAT1*), fatty acid elongase (*YLR372W*), and enoyl-CoA hydratase (*PHS1*) genes enhance the hydrolysis of fatty acids to acetyl-CoA and decrease fatty acid synthesis from acetyl-CoA. Upregulation of aldehyde dehydrogenase (*ALD6*) and bifunctional alcohol dehydrogenase (*SFA1*) genes promote the conversion of acetyl-CoA to ethanol. Upregulation of aryl-alcohol dehydrogenases (*AAD3* and *AAD4*),

alpha-trehalase (*NTH1*), hexokinase 1 (*HXK1*), *YJR096W*, glycerol 2-dehydrogenase (*GCY1*), and *ALD6* genes direct the flow of carbon into the glycolytic pathway, then into pyruvate and ethanol. The expression of alanine transaminase (*ALT2*), aspartate transaminase (*AAT1*), and *ALD6* genes promote the breakdown of amino acid carbon skeletons to form α-ketoglutaric acid and succinyl-CoA. Ethanol could be a major source of acetyl-CoA and NADPH indirectly during fermentation by *S. cerevisiae*, and *ALD6* plays an important role in this process [46]. Understanding the different metabolic processes is relevant in regard to engineered *S. cerevisiae* deployed for ethanol production.

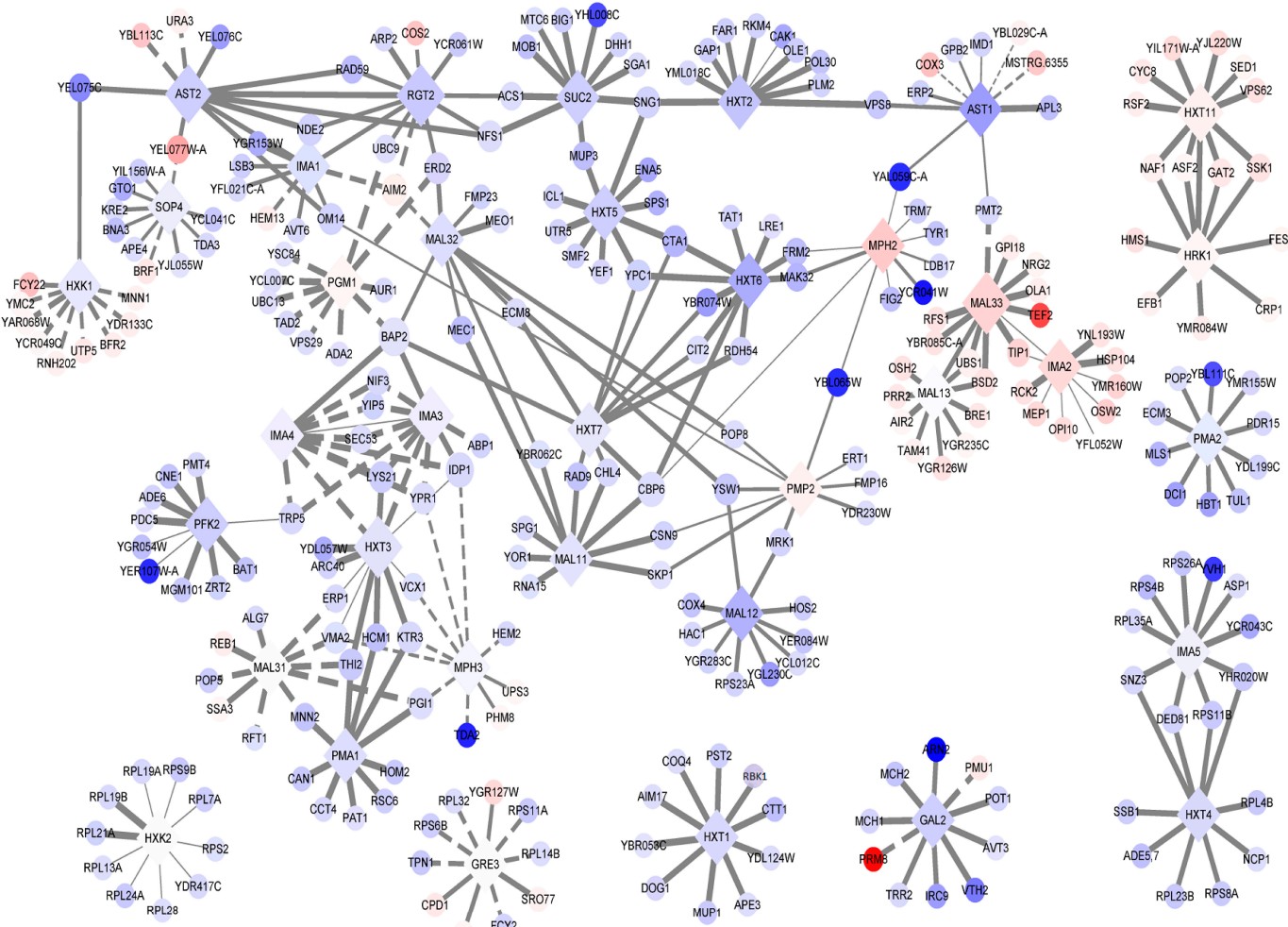

**Figure 12.** Protein–protein interaction network of sucrose consumes related genes. Core genes are shown in a diamond shape, and others in a circle, labeled with gene ID or gene symbol. The color is gradually changed according to log2FC; red represents log2FC > 0; the darker the red, the larger the upregulation ratio; blue represents log2FC < 0; the bluer the color, the larger the downregulation ratio. Positive correlations are shown by solid gray lines, and negative correlations are shown by dashed gray lines; the thickness of the line gradually changes according to the absolute value of the correlation coefficient; the larger value, the thicker line.

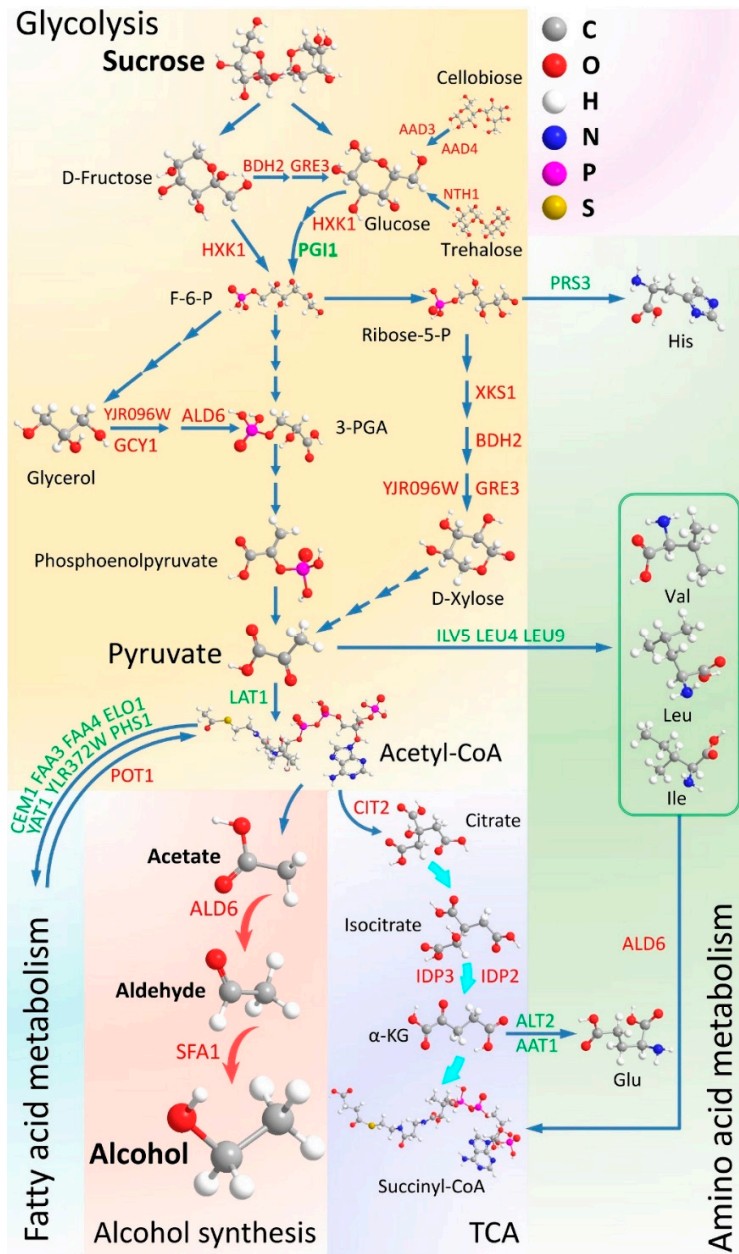

**Figure 13.** Exclusive genes in the MT strain are involved in carbon metabolism. Yellow box: glycolysis; blue box: fatty acid metabolism; green box: amino acid metabolism; purple box: tricarboxylic acid cycle; red box: alcohol biosynthesis. Red represents upregulated gene; green represents downregulated gene. The 3D structures of compounds were downloaded from PubChem (https://pubchem.ncbi.nlm.nih.gov/, accessed on 28 September 2022).

## 5. Conclusions

We obtained a mutant *S. cerevisiae* strain with improved capacity for ethanol fermentation and analyzed its genomic structure and gene expression changes. The SNPs had a high distribution density on all chromosomes except for the ends of chromosome I and II, while the InDels had the highest distribution density on the mitochondria genome. GO and KEGG enrichment for associated genes of SNPs and InDels were performed, and the significantly enriched metabolic pathway included carbohydrate metabolism, amino acid metabolism, metabolism of cofactors and vitamins, and lipid metabolism. There were significant differences in gene expression between the two strains during fermentation. The results of KEGG enrichment of DEGs between two strains were consistent with the result

of KEGG enrichment for associated genes of SNPs and InDels. Gene expression trends of the two strains were recorded on a timeline during fermentation. Venn diagram analysis revealed exclusive genes in the MT strain. KEGG enrichment analysis of these genes showed that genes involved in sugar metabolism, the MAPK pathway, and fatty acid and amino acid degradation were mainly upregulated. In contrast, genes involved in oxidative phosphorylation and ribosome, fatty acid, and amino acid biogenesis were mainly down-regulated. Protein interaction analysis of these genes showed that *PGI1*, *SPC3*, *PFK2,* and *GRE3* were the major hub genes in the network, linking sugar, amino acid, and fatty acid metabolism; the MAPK pathway; oxidative phosphorylation; ribosome biogenesis; protein export; and cellular senescence. This work provides a reference for the future construction of engineered strains of *S. cerevisiae* with excellent ethanol fermentation capacity.

**Supplementary Materials:** The following supporting information can be downloaded at: https://www.mdpi.com/article/10.3390/fermentation9050483/s1, Figure S1: Distribution of gene expression abundance; Figure S2: Heat map of the Pearson correlation coefficient R between samples; Table S1: Primers for qPCR; Table S2: Base information of genome sequencing before filter; Table S3: Base information of genome sequencing after filter; Table S4: Alignment to the reference genome; Table S5: Statistics of genome coverage; Table S6: Locations of SNPs on the genome; Table S7: Coding information of SNPs; Table S8: Location of InDels on the genome; Table S9: Coding information of InDels; Table S10: Hybrid information of SNPs and InDels; Table S11: Base information of RNA-Seq; Table S12: Alignment to the reference genome; Table S13: Location of alignment to the reference genome; Table S14: Gene number detection; Table S15: Gene coverage of each sample.

**Author Contributions:** Data curation, Z.L.; formal analysis, X.C.; funding acquisition, Y.C.; investigation, Q.L.; methodology, Y.W. and R.W.; project administration, Y.C.; resources, D.C.; software, Z.L.; writing—original draft, Y.C.; writing—review and editing, L.G. All authors have read and agreed to the published version of the manuscript.

**Funding:** This work was funded by the Natural Science Foundation of Guangxi, China (2020GXNS-FAA259021, 2022JJA130143, 2020GXNSFBA297070). The National Natural Science Foundation of China (32260018, 12264005). The Natural Science Foundation of Sichuan Province, China (2022NSFSC0723).

**Institutional Review Board Statement:** Not applicable.

**Informed Consent Statement:** Not applicable.

**Data Availability Statement:** The raw reads of WGS and RNA-Seq were uploaded to the NCBI (SRA) database with the accession number PRJNA885247. (https://www.ncbi.nlm.nih.gov/sra/?term=PRJNA885247, accessed on 28 February 2022).

**Conflicts of Interest:** The authors declare that the research was conducted in the absence of any commercial or financial relationships that could be construed as a potential conflict of interest.

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
