# Peer review of "Omics Sequencing of Saccharomyces cerevisiae Strain with Improved Capacity for Ethanol Production"

_fermentation, doi:10.3390/fermentation9050483_

Round 1

Reviewer 1 Report

This is a manuscript review for ''Omics sequencing of Saccharomyces cerevisiae with high capacity for ethanol production''

The manuscript brings new insights and is well-written. The issues below should be addressed

Abstract: reduce to about 200 words as per the Journal's guide

Line 380: Replace 'shown' with 'show'

Wild-type strain source: The source is still unclear. Reference was made to a previous publication. In that publication, it was just referred to as an ethanol-yielding strain. Authors should state the original source of the strain

What is the limitation of the study? Surely, findings from two strains may not be enough to generalise or justify the ample omics analysis carried out.

Abbreviations used in some figures should be described in full in the legend. e.g. Figure 3

The high-capacity ethanol-producing mutant is just about 3% higher than the wild type. Authors should compare ethanol production in other high-producing strains to justify the 'high-producing tag.

as indicated above

Reviewer 2 Report

The summary and conclusion section should be rewritten to better describe the findings.

In the caption to Fig. 1 it is necessary to give the composition of the medium.

The authors do not comment in any way on the fact that growth stops by 20 h of cultivation. Although there is a high concentration of reducing sugars in the medium, possibly glucose. Is the cessation of growth associated with a large production of ethanol, which inhibits growth?

It would be necessary to determine the glucose content in the medium during cultivation, for example, by the standard enzymatic method.

The difference in ethanol production between strains is small. Since in fig. 1A, the growth rates of the strains are close, it is necessary to describe in detail how the mutant strain was screened.

It should be explained why the cultivation was carried out at a reduced level of aeration (cultivation at 180 rpm) - so that the cells do not consume the ethanol they produce or for some other purpose.

In the discussion, it should be noted which genes the authors consider responsible for the observed increase in ethanol production.

Small remarks

Title:

Saccharomyces cerevisiae strain

Abstract:

distribution density in mitochondria genome

The physiological parameters of high capacity for ethanol fermentation need to be indicated in Abstract

Round 2

Reviewer 2 Report

My suggestion is  - accept